# Differences in the Glenohumeral Joint before and after Unilateral Breast Cancer Surgery: Motion Capture Analysis

**DOI:** 10.3390/healthcare10040707

**Published:** 2022-04-11

**Authors:** Silvia Beatríz García-González, María Raquel Huerta-Franco, Israel Miguel-Andrés, José de Jesús Mayagoitia-Vázquez, Miguel León-Rodríguez, Karla Barrera-Beltrán, Gilberto Espinoza-Macías

**Affiliations:** 1Departamento de Ciencias Aplicadas al Trabajo, Universidad de Guanajuato, Campus Leon, Guanajuato 37128, Mexico; betty_garcia.27@hotmail.com (S.B.G.-G.); mrhuertafranco@ugto.mx (M.R.H.-F.); 2Rehabilitación, Hospital Regional de Alta Especialidad del Bajío, Leon 37660, Mexico; karlabarrerabeltran@yahoo.com.mx; 3Biomecánica, Centro de Innovación Aplicada en Tecnologías Competitivas, Leon 37545, Mexico; jmayagoitia@ciatec.mx; 4Maestría en Tecnología Avanzada, Universidad Politécnica de Guanajuato, Cortazar 38496, Mexico; migueleon@upgto.edu.mx; 5Departamento de Oncología, Hospital General de Leon, Guanajuato 37672, Mexico; drgilespinoza@gmail.com

**Keywords:** kinematics, mastectomy, breast cancer, glenohumeral joint

## Abstract

After mastectomy, women might lose mobility and develop kinematic changes in the shoulder. The objective of this research was to compare the kinematics of the glenohumeral joint in women, before and after unilateral breast cancer surgery. This was a longitudinal study with a pre- and post-evaluation design; in total, 15 Mexican women who had a mastectomy for breast cancer and who received a physical therapy program after surgery were evaluated. Flexion–extension and abduction–adduction movements of the glenohumeral joint were evaluated (15 days before and 60 days after mastectomy). For the kinematic analysis of the glenohumeral joint, an optoelectronic motion capture system was used to monitor 41 reflective markers located in anatomical landmarks. There was no significant difference in the range of motion of the glenohumeral joint when comparing pre- and post-mastectomy, flexion–extension (*p* = 0.138), and abduction–adduction (*p* = 0.058). Furthermore, patients who received chemotherapy (53%) before mastectomy were more affected (lower range of motion) than those who did not receive it. There were no significant differences in the kinematics of the glenohumeral joint after mastectomy in this group of patients who received a physical therapy program after surgery. Moreover, patients who received chemotherapy treatment before breast cancer surgery tended to have a lower range of motion than those who did not receive it. Therefore, it is necessary for the physical rehabilitation team to attend to these patients even before the mastectomy.

## 1. Introduction

Breast cancer is a compilation of malignant tumors located in the mammary glands, which originates from the uncontrolled growth of malignant cells [1,2]. Worldwide, breast cancer represents the most diagnosed cancer in women, with an incidence for the year 2020 of 2.26 million new cases [3,4]. The breast cancer trends in Mexico are not promising, as they presented a constant increment since the year 2000. Since then, breast cancer has been considered one of the main causes of death in women [5]. In 2020, Mexico reported 29,929 new cases of people with breast cancer, which represented 15.3% of diagnosed cancers [6]. There are different ways to treat breast cancer. The treatment depends on the stage of cancer—it could be a surgical procedure, radiation therapy, chemotherapy, hormonal therapy, and targeted therapy [7]. Although medicine has considerably improved, the treatments applied to breast cancer patients still leave adverse effects. Women who receive treatment for breast cancer have impaired functionality, which affects their quality of life [8,9]. Mastectomy could affect the upper limbs, with persistent pain, lymphedema, weakness, and restriction of shoulder mobility [10,11,12]. In addition, it has been shown that, after breast cancer surgery, women present alterations in the kinematics of the upper limbs [13,14,15,16].

Kinematics provides objective information to evaluate and understand musculoskeletal pathologies, allowing for the improvement of therapeutic treatments [17,18]. However, these kinematic evaluations require complex equipment (motion capture systems) that are often difficult to access and manage. Therefore, evaluations of the upper extremity in patients with mastectomy using these tools are limited [15,16]. In addition, the studies found in the literature evaluate the kinematics in the postoperative period and compare it with a control group or with the unaffected arm [13,15,16,19], without considering a baseline measurement (before mastectomy). Consequently, the kinematic changes that women present in the glenohumeral joint before and after mastectomy are not clearly understood. There are studies in the literature, in which different methodologies have been used to measure the kinematics of the glenohumeral joint, in women who underwent surgery for breast cancer; however, their results have shown differences in the ranges of motion before and after mastectomy. For example, Min et al. found significant differences when comparing the range of motion pre-mastectomy and four weeks post-mastectomy. For the flexion–extension movements, it was a difference of 32 degrees, and for the abduction–adduction movements, it was 77 degrees [20]. Flores et al. reported differences when comparing measurements before and 2–3 weeks after breast cancer surgery; for the flexion–extension movements, mean values were 151 vs. 105 degrees, and for the abduction–adduction movements, mean values were 144 vs. 83 degrees [21]. Although there are some studies that evaluate the shoulder before and after breast cancer surgery [20,21,22], the glenohumeral kinematics still need to be studied at an early stage using vision equipment (non-invasive motion measurement system). It is imperative to develop studies in a shorter period of time after surgery to strengthen the lack of knowledge in the scientific research literature. A short-term kinematic assessment of the shoulder before and after breast cancer surgery will help to understand the recovery process of patients. Furthermore, it is important to analyze the effect of a physical therapy treatment used in women after surgical intervention [23,24,25,26].

Therefore, the objective of this research was to compare the kinematics of the glenohumeral joint (flexion–extension and abduction–adduction movements), 15 days before and 60 days after breast cancer surgery, in women who received physical therapy after surgery.

## 2. Materials and Methods

### 2.1. Anthropometric Data of the Patients

In total, 15 right-handed Mexican females, with a mean age of 46.7 ± 8.2 years and a mean body weight of 68.4 ± 7.2 kg, participated in the study. Table 1 shows the anthropometric and clinical information of the patients. The sample size was calculated using Epidat 3.1 software (version 3.1, Consellería de Sanidade, Xunta de Galicia, Santiago de Compostela, España). The statistical power considered in the study was 80%, and the confidence interval was 95%. The mean and standard deviation of the range of motion were estimated from a previous study [19]. Eight women (53.3%) had a mastectomy on the right side and seven (46.6%) on the left side.

This was an observational, longitudinal, study with pre- and post-assessment. The inclusion criteria were women between 30 and 60 years of age, scheduled for unilateral mastectomy for breast cancer, who completed the pre- and post-kinematic evaluations, and who also received a physiotherapeutic exercise program immediately after mastectomy. The physiotherapeutic program consisted of active and progressive mobility exercises of the upper limbs during the following two months after surgery [23]. Furthermore, the evaluation post-surgery was performed considering the removal of the drain and complete healing of the surgical wound.

Women who did not completely heal the surgical wound before the 60 days, did not sign informed consent, or did not assist in the second evaluation after the surgery were excluded from the research.

This study was carried out following the ethical principles dictated by the Declaration of Helsinki, for the treatment of human beings in clinical intervention studies [27]. The project was explained to the participants, and all of them signed informed consent before carrying out the evaluations. The research protocol was approved by the ethics and research committee of Hospital Regional de Alta Especialidad del Bajío (CI/HRAEB/009/2020) and Hospital General de León (PIHGL-CEIS-016-2021).

### 2.2. Experimental Protocol

The evaluations were carried out at the Biomechanics Laboratory of Centro de Innovación Aplicada en Tecnologías Competitivas; all participants were evaluated 15 days before and 60 days after the mastectomy. The glenohumeral joint kinematics was measured with reflective markers and seven infrared cameras of the Vicon System (Nexus version 2.8.1.111866h x86, Vicon Motion System Ltd., Oxford, UK), at a sampling frequency of 100 Hz. First, the setup of the Vicon System was performed. Then, 39 spherical reflective markers (passive markers, 10 mm diameter) were attached to the participant. The markers were placed in specific bony landmarks following the Vicon manual and using double-sided tape, as shown in Figure 1. Furthermore, two extra markers were placed in the medial epicondyle of the left and right arms. The medial epicondyle markers were used to create the orthogonal coordinate systems of each arm. To ensure repeatability between testing sessions, the experimental protocol was performed by the same investigator for both sessions (before and after mastectomy).

The flexion–extension and abduction–adduction movements of the affected arm (body side where the mastectomy was performed) were analyzed. Before starting each test, the participants were explained how the movement should be executed, and time was given to familiarize themselves with the equipment and experimental protocol.

Five trials of unilateral flexion–extension and abduction–adduction of the arms were performed. The execution time for each trial was 20 s for 10 cycles paced using a metronome. Each cycle was performed in a period of time of 2 s. The participants started the movement in a neutral position lifting the upper limb and going backward to the beginning. Participants took a break of three minutes among trials to rest and avoid fatigue.

### 2.3. Data Processing

Once the evaluations were completed, data processing continued in the Vicon Nexus software (Nexus version 2.8.1.111866h x86), where the 41 reflective markers placed on the participants were labeled. Next, the data were exported to MATLAB R2015a software (version 8.5.0.197613, The MathWorks Inc., Natick, MA, USA), in which three orthogonal coordinate axes systems were created (Figure 2a). The first coordinate system (CS) was created with markers of the torso (the mathematical procedure is explained through Equations (1)–(6), the second one was created with markers of the left arm (Equations (7)–(12)), and the last one was created with markers of the right arm (similarly to the CS of the left arm).

The coordinate system for the torso was defined as follows:(1)$${\vec{U}}_{1}\left(\hat{i},\hat{j},\hat{k}\right)=\frac{\left(C7\;-\;T10\right)}{\left|C7\;-\;T10\right|}$$
(2)$${\vec{U}}_{2}\left(\hat{i},\hat{j},\hat{k}\right)=\frac{\left(STRN\;-\;T10\right)}{\left|STRN\;-\;T10\right|}$$
(3)$${\vec{X}}_{T}\left(\hat{i},\hat{j},\hat{k}\right)\;=\;{\vec{U}}_{1}\;\times \;{\vec{U}}_{2}$$
(4)$${\vec{Y}}_{T}\left(\hat{i},\hat{j},\hat{k}\right)\;=\;{\vec{X}}_{T}\;\times \;{\vec{U}}_{1}$$
(5)$${\vec{Z}}_{T}\left(\hat{i},\hat{j},\hat{k}\right)\;=\;{\vec{Y}}_{T}\;\times \;{\vec{X}}_{T}$$
(6)$$R_{T}\;=\;\left[\begin{array}{ccc} {\vec{X}}_{T}\left(\hat{i}\right) & {\vec{X}}_{T}\left(\hat{j}\right) & {\vec{X}}_{T}\left(\hat{k}\right)\\ {\vec{Y}}_{T}\left(\hat{i}\right) & {\vec{Y}}_{T}\left(\hat{j}\right) & {\vec{Y}}_{T}\left(\hat{k}\right)\\ {\vec{Z}}_{T}\left(\hat{i}\right) & {\vec{Z}}_{T}\left(\hat{j}\right) & {\vec{Z}}_{T}\left(\hat{k}\right) \end{array}\right]$$
where ${\vec{U}}_{1}$ and ${\vec{U}}_{2}$ are the unit vectors in the sagittal plane created from markers of the torso (vertebra C7, vertebra T10, and sternum marker STRN); ${\vec{X}}_{T}$ is the unit vector orthogonal to the sagittal plane; ${\vec{Y}}_{T}$ is the orthogonal unit vector created from the cross product between the ${\vec{X}}_{T}$ and ${\vec{U}}_{1}$ vectors; ${\vec{Z}}_{T}$ is the orthogonal unit vector of the torso created from the cross product between the ${\vec{Y}}_{T}$ and ${\vec{X}}_{T}$ vectors; $R_{T}$ is the 3 × 3 matrix of the coordinate system of the torso (*T*).

The coordinate system for the left arm was determined as follows:(7)$${\vec{V}}_{1}\left(\hat{i},\hat{j},\hat{k}\right)\;=\;\frac{\left(L\;AC\;joint\;-\;LELB\right)}{\left|L\;AC\;joint\;-\;LELB\right|}$$
(8)$${\vec{V}}_{2}\left(\hat{i},\hat{j},\hat{k}\right)\;=\;\frac{\left(L\;Epicondyle\;-\;LELB\right)}{\left|L\;Epicondyle\;-\;LELB\right|}$$
(9)$${\vec{y}}_{LA}\left(\hat{i},\hat{j},\hat{k}\right)\;=\;{\vec{V}}_{1}\;\times \;{\vec{V}}_{2}$$
(10)$${\vec{x}}_{LA}\left(\hat{i},\hat{j},\hat{k}\right)\;=\;{\vec{V}}_{1}\;\times \;{\vec{y}}_{LA}$$
(11)$${\vec{z}}_{LA}\left(\hat{i},\hat{j},\hat{k}\right)\;=\;{\vec{y}}_{LA}\;\times \;{\vec{x}}_{LA}$$
(12)$$R_{LA}\;=\;\left[\begin{array}{ccc} {\vec{x}}_{LA}\left(\hat{i}\right) & {\vec{x}}_{LA}\left(\hat{j}\right) & {\vec{x}}_{LA}\left(\hat{k}\right)\\ {\vec{y}}_{LA}\left(\hat{i}\right) & {\vec{y}}_{LA}\left(\hat{j}\right) & {\vec{y}}_{LA}\left(\hat{k}\right)\\ {\vec{z}}_{LA}\left(\hat{i}\right) & {\vec{z}}_{LA}\left(\hat{j}\right) & {\vec{z}}_{LA}\left(\hat{k}\right) \end{array}\right]$$
where ${\vec{V}}_{1}$ is the unit vector longitudinally to the arm and created from the markers of the lateral elbow (LELB) and the left acromioclavicular joint (L AC joint); ${\vec{V}}_{2}$ is the unit vector created from the lateral (LELB) and medial epicondyles (L Epicondyle); ${\vec{y}}_{LA}$ is the orthogonal unit vector created from the cross product of the ${\vec{V}}_{1}$ and ${\vec{V}}_{2}$ vectors; ${\vec{x}}_{LA}$ is the orthogonal unit vector created from the cross product of the ${\vec{V}}_{1}$ and ${\vec{y}}_{LA}$ vectors; ${\vec{z}}_{LA}$ is the orthogonal unit vector created from the cross product of the ${\vec{y}}_{LA}$ and ${\vec{x}}_{LA}$ vectors; $R_{LA}$ is the 3 × 3 matrix of the coordinate system of the left arm (LA).

The relative movement of the left arm with respect to the torso was described from the following mathematical expression (Equation (13)):(13)$$R_{LA}^{T}\;=\;R_{T}R_{LA}^{-1}$$
where $R_{LA}^{T}$ is the 3 × 3 matrix which describes the relative movement of the left arm with respect to the torso; $R_{LA}^{-1}$ is the inverse matrix of the left arm.

The analysis of data was performed on the affected side and compared before and after the surgery. Of the five trials recorded, the data for the first and last trials in each participant were excluded. Then, an average of the repetitions was performed before and after mastectomy. For this investigation, the range of motion of the glenohumeral joint was calculated from the vertical axis (${\vec{z}}_{LA}$ axis) of the affected arm with respect to the vertical axis (${\vec{Z}}_{T}$ axis) of the torso, as shown in Figure 2b.

### 2.4. Statistical Analysis

The statistical analysis of the data was performed in the software IBM SPSS statistics, version 25 (IBM corporation, Armonk, NY, USA). The normality of the data was analyzed with the Shapiro–Wilk test. To test the hypothesis that there was a significant difference in the kinematics of the glenohumeral joint before and after mastectomy, Student’s *t*-test for paired samples or a Wilcoxon rank-sum test was applied. It was established that the difference was significant when the *p*-value was less than 0.05.

## 3. Results

From the kinematic evaluations, a comparison of the results of the participants before and after mastectomy was made.

Figure 3 shows the kinematics of the vertical axis (${\vec{z}}_{LA}$ axis) of the affected arm with respect to the vertical axis of the torso (${\vec{Z}}_{T}$ axis). Moreover, Figure 3 shows the trajectory of the glenohumeral joint of the affected arm during the flexion–extension movement before and after mastectomy. In Figure 3a, the mean and standard deviation of the 15 participants before the breast cancer surgery are presented, while Figure 3b shows the mean and standard deviation of the same participants after mastectomy.

Before the surgery, the mean range of motion was 103.5 ± 14.7 degrees, and after the surgery, the average was 96.8 ± 17.6 degrees, respectively. There was a small reduction in the range of motion after the breast cancer surgery. However, after applying Student’s *t*-test for paired samples, no significant differences were found between both kinematic evaluations (*p* = 0.138).

In Figure 4, the kinematics of the glenohumeral joint of the affected arm during the abduction–adduction movement, before and after mastectomy, is presented. An average range of motion of 106.5 ± 20.5 degrees was obtained before breast cancer surgery (Figure 4a), and 96.4 ± 16.9 degrees after mastectomy, respectively (Figure 4b). Although there was a reduction in the range of motion after the surgery, the results of Student’s *t*-test for paired samples did not show a significant difference for both kinematic evaluations (*p* = 0.058). However, the results are shown at the limit of statistical significance.

The impact of breast cancer surgery on the kinematics of the glenohumeral joint will depend on the type of surgery, lymph nodes extracted, or any other adjuvant therapy. From the physiological point of view, the emotional state of patients diagnosed with breast cancer can be affected. Moreover, breast cancer surgery can reduce the free movement of the upper limbs by the retraction of the soft tissue during the healing of the wound.

In addition, musculoskeletal nociceptive pain arises from the fact that treatments for breast cancer directly involve the neuromusculoskeletal tissues of one or more limb areas, causing shoulder pain, joint limitation, and hypoesthesia [28].

In addition, chemotherapy is a treatment that uses powerful chemical drugs to kill cancer cells. However, it can produce side effects on the patients; most of the time, it affects their physical condition, appearance, and quality of life. Breast cancer patients undergoing adjuvant chemotherapy reduce their daily energy expenditure during therapy, which is associated with a loss of muscle mass. Furthermore, it was shown that skeletal muscle status is of clinical relevance because it is associated with treatment complications and time-to-tumor progression [28]. Therefore, a further analysis was performed among patients who received chemotherapy treatment before mastectomy (*n* = 8), and those who did not receive it (*n* = 7). This was carried out with the purpose of identifying the effect of the treatment on the kinematics of the glenohumeral joint prior to the surgery. However, the comparison between these two groups shows a significant difference between the flexion–extension movement (*p* = 0.001), and the abduction–adduction movement, respectively (*p* = 0.015). The results of this analysis are shown in Table 2.

Based on the previous analysis (patients with and without chemotherapy treatment), a new classification of two groups was performed to identify the effect of the treatment on the kinematics of the glenohumeral joint before and after mastectomy. Group 1 involved women who did not receive the treatment, and Group 2 women who received it. No significant differences were found in the flexion–extension movement for the patients who did not receive chemotherapy treatment before mastectomy (*p* = 0.384), nor in the group of patients who received this treatment (*p* = 0.243). Similarly, as regards the abduction–adduction movement, no significant differences were found in both groups. In the group that did not receive chemotherapy, the *p*-value was 0.233, and in the group that received chemotherapy, the *p*-value was 0.176. Table 3 shows in more detail the results of the statistical analysis.

## 4. Discussion

The kinematics of the glenohumeral joint (of women whose arm was affected with breast cancer and received a therapeutic treatment after surgery) were compared before and after mastectomy. No significant difference was found for the flexion–extension movement (*p* = 0.138). Similarly, the abduction–adduction movement did not present a significant difference when comparing the kinematic evaluations (*p* = 0.058). No previous studies have been found that compare the kinematics of the glenohumeral joint, before and after mastectomy, using an optoelectronic motion capture system with the same methodology implemented in our study.

In the studies developed by Min et al. [20] and Flores et al. [21], the measurements after mastectomy were taken between the second and fourth week after surgery. The differences in the results could be originated due to the variations in the times in which the post-surgical evaluations were carried out. In the present research, the evaluation time was eight weeks after mastectomy. In addition, the studies mentioned above [20,21] do not specify whether the participants received physical therapy after surgery. Testa et al. demonstrated that women who received an immediate physical rehabilitation program after breast cancer surgery improved glenohumeral joint mobility [29]. Similarly, it has been found that exercise therapy promotes normal motor control and decreases disability [30,31]. Moreover, it has been reported that women who are physically more active tend to have a reduced risk of breast cancer death, compared with sedentary women [32].

In the current investigation, it was observed that the abduction–adduction movement had a higher affectation than the flexion–extension movement. This agrees with the studies by the authors mentioned above, in which major differences were found in the abduction–adduction movement before and after the surgery [20,21]. This might be caused by skin retraction during healing after mastectomy, in addition to pectoral muscle shortening as a protective mechanism [33,34].

The results of the analysis of the patients who received the chemotherapy treatment and those who did not receive it before breast cancer surgery demonstrate a significant difference in the flexion–extension (*p* = 0.001) and the abduction–adduction movements (0.015). This means that patients with the chemotherapy treatment present a significantly lower range of motion. As explained by Shamley et al., there is a significant association between chemotherapy and alterations in joint movement patterns [35]. These differences could be attributed to various factors—namely, chemotherapy is an extensive treatment, it has repercussions on work absence [36], physical functions are reduced, and generates weakness and fatigue [37]. Moreover, the presence of peripheral neuropathy induced by chemotherapy affects sensory functions and produces motor symptoms manifested as tingling, numbness, and neuropathic pain [38]. Furthermore, from the diagnosis of breast cancer and the start of treatment, it affects the emotional and social state of patients, as well as their body image [39]. All of these factors could influence the results of the kinematics.

Furthermore, it was noticed that the average age of the women evaluated in this study was 46.7 ± 8.2 years, which is similar to that reported by other authors who have studied breast cancer. These investigators have reported an average age of 52 ± 12.1 years, which falls in the age group of 41 to 50 years, one of the most affected [19,40,41]. Another important aspect observed in this investigation was the average body mass index (BMI) of 27.8 ± 3.1 kg/m^2^ so most of the participants were overweight, according to the values established by the World Health Organization (WHO). In other studies, it has been shown that 38–41% of women with breast cancer were overweight [40,42]. Moreover, obesity is a frequent health problem in these patients; however, in this investigation, women with a BMI higher than 34 kg/m^2^ were not included in the study, as the skinfold might produce interference during kinematic evaluation.

To our knowledge, this is the first study to measure the kinematics of the glenohumeral joint in a female Mexican population with an optoelectronic system before and after mastectomy. Furthermore, from the results found in the study, it can be recommended that the recovery process of the patients after breast cancer surgery should be supported by exercise therapy of the upper limbs.

The shoulder has been considered one of the most complex joints of the upper limbs, so understanding its kinematics is difficult. Our research contributes to an easy understanding of the relative movement of the arm with respect to the trunk, making practical the analysis of the patients who have this type of issue. In clinical practice, our methodology helps the physical rehabilitation team to understand the recovery process of patients with breast cancer.

However, our study presents some limitations: first, the relatively small sample size does not allow the generalization of the outcomes; second, the study evaluated movements of the arms mainly in the sagittal and frontal planes. Future studies should consider the assessment of complex movements of the arms during daily life activities. Moreover, further investigations need to be carried out by considering more variables related to breast cancer surgery, such as the type of surgery, the number of lymph nodes extracted, or the chemotherapy treatment. Although the study presents some limitations, its results contribute to understanding the kinematics of the glenohumeral joint before and after mastectomy.

## 5. Conclusions

This investigation presented a methodology of practical use (non-invasive) to evaluate the range of motion of the glenohumeral joint before and after mastectomy. Furthermore, the results of this study contribute to understanding the kinematics of the glenohumeral joint and the recovery process of patients who had breast cancer surgery. Although there was a reduction in the range of motion of the glenohumeral joint after mastectomy, no significant differences were found in the flexion–extension and abduction–adduction movements in this group of patients who received physical therapy after mastectomy. In addition, patients who received chemotherapy treatment before breast cancer surgery tended to have a lower range of motion. Therefore, it is necessary for the physical rehabilitation team to attend to these patients even before mastectomy.

## Figures and Tables

**Figure 1 healthcare-10-00707-f001:**
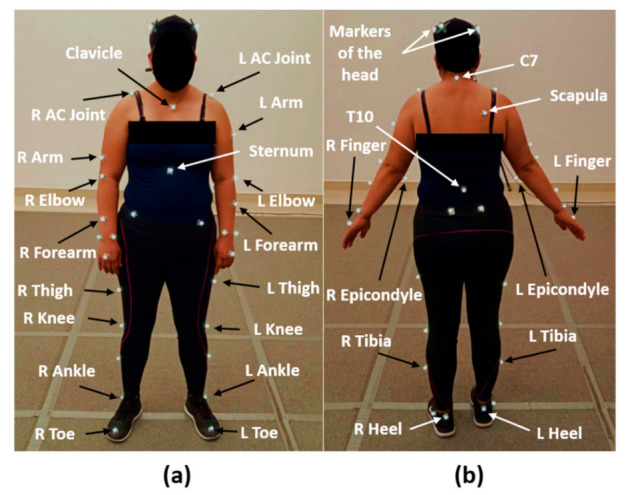
Placement of the markers in specific bony landmarks: (**a**) anterior view and (**b**) posterior view. R and L, the right and left sides of the evaluated volunteer.

**Figure 2 healthcare-10-00707-f002:**
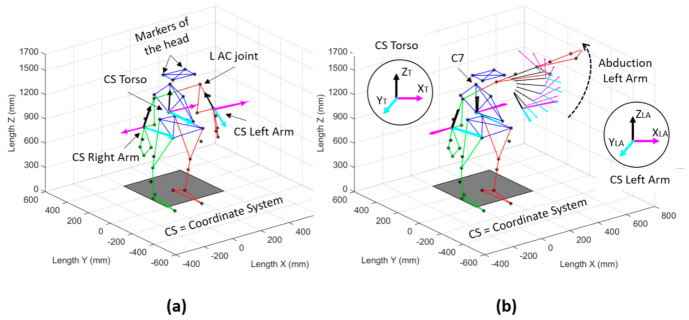
Three orthogonal coordinate systems were created to measure the kinematics of the glenohumeral joint of the affected side: (**a**) isometric view of the reconstructed markers of the participant; (**b**) abduction movement of the left arm. CS, coordinate system; L AC joint, left acromioclavicular joint; C7, 7th cervical vertebra.

**Figure 3 healthcare-10-00707-f003:**
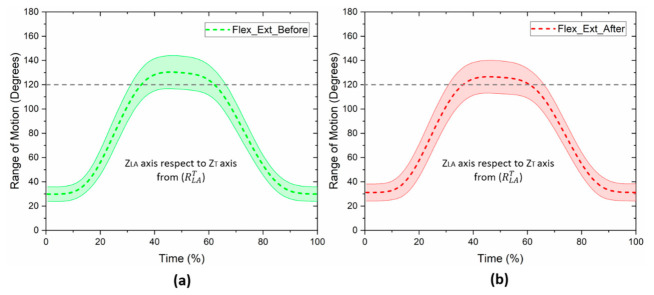
Mean and standard deviation of the range of motion of the glenohumeral joint, before and after breast cancer surgery: (**a**) flexion–extension before mastectomy; (**b**) flexion–extension after mastectomy. Movement of the ${\vec{z}}_{LA}$ axis of the arm with respect to the ${\vec{Z}}_{T}$ axis of the torso, from $R_{LA}^{T}$.

**Figure 4 healthcare-10-00707-f004:**
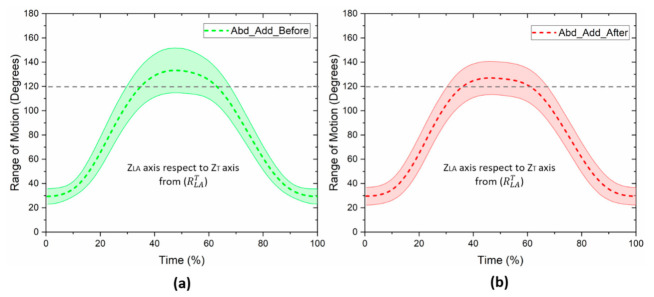
Mean and standard deviation of the range of motion of the glenohumeral joint, before and after breast cancer surgery: (**a**) abduction–adduction before mastectomy; (**b**) abduction–adduction after mastectomy. Movement of the ${\vec{z}}_{LA}$ axis of the arm with respect to the ${\vec{Z}}_{T}$ axis of the torso, from $R_{LA}^{T}$.

**Table 1 healthcare-10-00707-t001:** Anthropometric characteristics of the participants and data related to breast cancer.

Anthropometric Data	Measurement Mean ± Std	Clinical Data	Measurement *n* (%)
Age (years)	46.7 ± 8.2	Stage cancer I and II	8 (53.3%)
Body weight (kg)	68.4 ± 7.2	Stage cancer III	7 (46.7%)
Height (cm)	156.9 ± 5.1	Chemotherapy (yes)	8 (53.3%)
Body mass index BMI (kg/m^2^)	27.8 ± 3.1	Axillary lymph node dissection (yes)	11 (73.3%)

std = standard deviation of the mean; *n* = the number of participants.

**Table 2 healthcare-10-00707-t002:** Comparison of the range of motion of the glenohumeral joint, between patients who received chemotherapy and those who did not receive this treatment before mastectomy.

Range of Motion (Degrees)	Patients with Chemotherapy Treatment (*n* = 8)	Patients without Chemotherapy Treatment (*n* = 7)	*p*-Value
Flexion–extension Mean ± std	92.0 ± 8.5	113.5 ± 11.3	0.001 *
Abduction–adduction Median (Q1–Q3)	101.2 (70.7–103.7)	114.8 (103.3–126.7)	0.015 **

Q1 and Q3, quartiles 1 and 3; std, standard deviation of the mean. * Significance was obtained with Student’s *t*-test for independent samples. ** Significance was obtained with the Mann–Whitney U test for independent samples.

**Table 3 healthcare-10-00707-t003:** Comparison of the range of motion of the glenohumeral joint before and after mastectomy. Group 1, women who did not receive chemotherapy before mastectomy; Group 2, women who received chemotherapy before mastectomy.

Group	Movement	Before (Degrees)	After (Degrees)	*p*-Value
1	Flexion–extension Mean ± std	113.5 ± 11.3	107.3 ± 15.4	0.384 *
2	Flexion–extension Mean ± std	92.0 ± 8.5	84.7 ± 11.3	0.243 *
1	Abduction–adduction Median (Q1–Q3)	114.8 (103.3–126.7)	108.4 (94.3–117.4)	0.233 **
2	Abduction–adduction Median (Q1–Q3)	101.2 (77.8–103.7)	87.6 (68.3–99.8)	0.176 **

Q1 and Q3, quartiles 1 and 3; std, standard deviation of the mean. * Significance was obtained with Student’s *t*-test for dependent samples. ** Significance was obtained with the Wilcoxon rank-sum test.

## Data Availability

The data in the current study are available on request from the corresponding author. The data are not publicly available due to ethical considerations.

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
