# Peer review of "Differences in the Glenohumeral Joint before and after Unilateral Breast Cancer Surgery: Motion Capture Analysis"

_healthcare, 2022, doi:10.3390/healthcare10040707_

Round 1
Reviewer 1 Report
Authors of the manuscript “Differences in the Glenohumeral Joint before and after Unilateral Breast Cancer Surgery: Motion Capture Analysis” raises a very topical issue in an affordable assessing the postoperative condition of female patients subjected to mastectomy. In more detail the paper reports how breast surgery and chemotherapy affect flexion-extension and abduction-adduction movements of the glenohumeral joint on the side of the operated breast. The data was collected with infrared cameras using reflective markers fixed on the patient’s body (motion capture analysis) and then processed in three coordinate systems. Despite there is a lack of scientific explanation regarding the observed effects revealed by the authors the paper is in a good agreement with the scope of Special Issue "Breast Cancer in Healthcare" and can be recommended for publication after major revision.
Remark 1.
The paper reports the results of study performed for the Mexican patients. If the authors follow an idea that their research can be prevalently used for the Mexican women considering differences in climate, food, etc. affecting females from different countries, this would be reasonable to specify this in the title of the paper (e.g. “…Breast Cancer Surgery of Mexican Females…”). If the authors do not want follow such an idea, this would be correct to mention that 15 females were from Mexican hospital, but to avoid highlighting this in the further text. This is hard to conclude that the results are appropriate just for Mexican patients and vice versa.
Remark 2.
Introduction is recommended to be enriched with reasoning why the authors selected the periods of 15 days before and 60 days after surgery. As a rule, recovery period is much longer than 2 months. Wasn’t is reasonable to wait for at least 6 months?
Remark 3.
Evaluation of mean age is confusing. In line 68 this is 46.7±8.2 years while in line 76 – between 30 and 60 years. Please clarify.
Remark 4.
Was any evaluation made for the control group of healthy women? This should be done.
Remark 5.
The patients’ morphology (e.g. cancer stage, patient age, proportion of tissue removed, chemotherapy strength and duration, etc.) should be presented to provide a comprehensive research. The authors should explain the results in scientific terms and find interplaying the motion restrictions and patient morphology. They should suggest how biochemical changes in human body induced by breast cancer (considering the patients morphology) influence limbs, in particular joints.
Remark 6.
Two paragraphs (lines 262-282) are recommended to be transferred into introduction section despite the next paragraph refers to them.
Remark 7.
The authors often mention that chemotherapy affected ability to move limb but no explanation why. This would be very fruitful to find any related suggestion in sections Discussion and Conclusions. In addition, the last section is recommended to be enriched with outlook on the method used in the paper, its perspectives and implementation in clinical practice.
Remark 8.
English grammar and style are good but please double-check a couple of misprints.
Author Response
30th/March/2022
Dear Dr. Georgios Iatrakis
Guest Editor Healthcare
Manuscript ID: healthcare-1652495
Thanks for reviewing the manuscript entitled “Differences in the Glenohumeral Joint before and after Unilateral Breast Cancer Surgery: Motion Capture Analysis”. We sincerely appreciate the commentaries that allow us to improve the article.
We have carefully attended to each commentary made, so we proceed to send the manuscript again for review. The observations made and the responses by the authors are described below. We hope that the modifications made will achieve the final approval by the evaluators and the editorial committee, thanks in advance.
Note. The changes in the manuscript are marked up using the “Track Changes” function of the Microsoft Word program.
Kind Regards,
*************************************************************************
Reviewer 1
*************************************************************************
Authors of the manuscript “Differences in the Glenohumeral Joint before and after Unilateral Breast Cancer Surgery: Motion Capture Analysis” raises a very topical issue in an affordable assessing the postoperative condition of female patients subjected to mastectomy. In more detail the paper reports how breast surgery and chemotherapy affect flexion-extension and abduction-adduction movements of the glenohumeral joint on the side of the operated breast. The data was collected with infrared cameras using reflective markers fixed on the patient’s body (motion capture analysis) and then processed in three coordinate systems. Despite there is a lack of scientific explanation regarding the observed effects revealed by the authors the paper is in a good agreement with the scope of Special Issue "Breast Cancer in Healthcare" and can be recommended for publication after major revision.
Thank you very much for reviewing our manuscript; we sincerely appreciate your commentaries. We have made some changes according to your suggestions to improve the quality of the document. Please find below the responses to each commentary made.
Remark 1.
The paper reports the results of study performed for the Mexican patients. If the authors follow an idea that their research can be prevalently used for the Mexican women considering differences in climate, food, etc. affecting females from different countries, this would be reasonable to specify this in the title of the paper (e.g. “…Breast Cancer Surgery of Mexican Females…”). If the authors do not want follow such an idea, this would be correct to mention that 15 females were from Mexican hospital, but to avoid highlighting this in the further text. This is hard to conclude that the results are appropriate just for Mexican patients and vice versa.
Thank you very much for your commentary. We have decided to specify just in the abstract, and the Materials and Methods section that the patients were Mexican females. Furthermore, we remove the word “Mexican” in other sections where it is not needed.
“15 Mexican women who had a mastectomy for breast cancer and who received…”, line 15.
“Fifteen right-handed Mexican females, mean age 46.7 ± 8.2 years, and…”, line 81.
Remark 2.
Introduction is recommended to be enriched with reasoning why the authors selected the periods of 15 days before and 60 days after surgery. As a rule, recovery period is much longer than 2 months. Wasn’t is reasonable to wait for at least 6 months?
Thanks for the commentary. We have selected the period of 15 days based on the programmed appointments of the patients with the hospital and the 60 days after the surgery considering the removal of the drain and complete healing of the surgical wound. According to the scientific literature, we noticed that most of the studies perform the evaluations in a longer period of time after the surgery [13,15,16]. However, others evaluated the patients in a shorter period of time (2-8 weeks) [20-22]. We identified this as an opportunity to strengthen the lack of knowledge in a shorter period of time considering the complete healing of the wound. To clarify this point we have added the following sentences in the introduction, and Materials and Methods sections.
“It is imperative to develop studies in a shorter period of time after surgery to strengthen the lack of knowledge in the scientific research literature”, line 69-71.
“Furthermore, the evaluation post-surgery was performed considering the removal of the drain and complete healing of the surgical wound”, lines 94-96.
[13] Crosbie J, Kilbreath SL, Dylke E, Refshauge KM, Nicholson LL, Beith JM, et al. Effects of mastectomy on shoulder and spinal kinematics during bilateral upper-limb movement. Phys Ther. 2010, 90, 679–92. https://doi.org/10.2522/ptj.20090104
[15] Brookham RL, Cudlip AC, Dickerson CR. Examining upper limb kinematics and dysfunction of breast cancer survivors in functional dynamic tasks. Clin Biomech. 2018, 55, 86–93. https://doi.org/10.1016/j.clinbiomech.2018.04.010
[16] Lang AE, Dickerson CR, Kim SY, Stobart J, Milosavljevic S. Impingement pain affects kinematics of breast cancer survivors in work-related functional tasks. Clin Biomech. 2019, 70, 223–30. https://doi.org/10.1016/j.clinbiomech.2019.10.001
[20] Min J, Kim JY, Yeon S, Ryu J, Min JJ, Park S, et al. Change in shoulder function in the early recovery phase after breast cancer surgery: A prospective observational study. J Clin Med. 2021, 10, 3416. https://doi.org/10.3390/jcm10153416
[21] Flores AM, Dwyer K. Shoulder impairment before breast cancer surgery. J Womens Health Phys Therap. 2014, 38, 118–24. https://doi.org/10.1097/jwh.0000000000000020
[22] Borstad JD, Szucs KA. Three-dimensional scapula kinematics and shoulder function examined before and after surgical treatment for breast cancer. Hum Mov Sci. 2012, 31, 408–18.
Remark 3.
Evaluation of mean age is confusing. In line 68 this is 46.7±8.2 years while in line 76 – between 30 and 60 years. Please clarify.
Thank you for noticing that. The value in line 68 refers to the mean and standard deviation of the age of the patients. On the other hand, the values in line 76 refer to the inclusion criteria to participate in the study. To clarify this, we have modified the sentence as follows:
“The inclusion criteria were women between 30 and 60 years of age, scheduled for unilateral mastectomy for breast cancer, completed the pre- and post-kinematic evaluations, and who also received a physiotherapeutic exercise program immediately after the mastectomy.”, lines 89-93.
Remark 4.
Was any evaluation made for the control group of healthy women? This should be done.
Thanks a lot for the commentary; we sincerely appreciate the suggestion. However, it is difficult to get a control group to compare our results before the submission date. Since the beginning, we planned the research as an observational study with a pre-and post-assessment design where the baseline measurement (before surgery) would be the control measurement.
Remark 5.
The patients’ morphology (e.g. cancer stage, patient age, proportion of tissue removed, chemotherapy strength and duration, etc.) should be presented to provide a comprehensive research. The authors should explain the results in scientific terms and find interplaying the motion restrictions and patient morphology. They should suggest how biochemical changes in human body induced by breast cancer (considering the patients morphology) influence limbs, in particular joints.
Thanks a lot for your commentaries. First, we have included a table with the anthropometric data of the participants, data of the breast cancer, and some features of the surgical procedure. This to be more specific about the patients included in the study. Then, a new subsection (2.1 Anthropometric data of the patients) was created in the Material and Methods section, where the information about the participants was included. The new table (Table 1) was added in that subsection.
Table 1. Anthropometric characteristics of the participants, and data related to breast cancer data.
|
Anthropometric data |
Measurement Mean ± std |
Clinical data |
Measurement n (%) |
|
Age (years) |
46.7 ± 8.2 |
Stage cancer I and II |
8 (53.3%) |
|
Body weight (kg) |
68.4 ± 7.2 |
Stage cancer III |
7 (46.7%) |
|
Height (cm) |
156.9 ± 5.1 |
Chemotherapy (yes) |
8 (53.3%) |
|
Body Mass Index BMI (kg/m2) |
27.8 ± 3.1 |
Axillary lymph node dissection (yes) |
11 (73.3%) |
Std = is the standard deviation of the mean; and n = represents the number of participants.
Moreover, a paragraph was added in the results section to improve the explanation of the motion restrictions and patient morphology after breast cancer surgery.
“The impact of the breast cancer surgery on the kinematics of the glenohumeral joint will depend on the type of surgery, lymph nodes extracted, or any other adjuvant therapy. From the physiological point of view, the emotional state of patients diagnosed with breast cancer can be affected. Moreover, breast cancer surgery can reduce the free movement of the upper limbs by retracting the soft tissue during the healing of the wound.”, lines 237-241.
“In addition, musculoskeletal nociceptive pain arises from the fact that treatments for breast cancer directly involve the neuromusculoskeletal tissues of one or more limb areas, causing shoulder pain, joint limitation, and hyposthenia [28].”, lines 242-244.
[28] Klassen O, Schmidt ME, Ulrich CM, Schneeweiss A, Potthoff K, Steindorf K, Wiskemann J. Muscle strength in breast cancer patients receiving different treatment regimes. J Cachexia Sarcopenia Muscle. 2017; 8(2): 305-316. https://doi.org/10.1002/jcsm.12165
Remark 6.
Two paragraphs (lines 262-282) are recommended to be transferred into introduction section despite the next paragraph refers to them.
Thank you very much for the suggestion. We have decided to move just the first paragraph (lines 262-271) to the introduction section (inserted between lines 57-66). The second paragraph (lines 274-283) was left in the discussion section in order to compare our results with the outcomes found by Min et al., and Flores et al. [20-21].
[20] Min J, Kim JY, Yeon S, Ryu J, Min JJ, Park S, et al. Change in shoulder function in the early recovery phase after breast cancer surgery: A prospective observational study. J Clin Med. 2021, 10, 3416. https://doi.org/10.3390/jcm10153416
[21] Flores AM, Dwyer K. Shoulder impairment before breast cancer surgery. J Womens Health Phys Therap. 2014, 38, 118–24. https://doi.org/10.1097/jwh.0000000000000020
Remark 7.
The authors often mention that chemotherapy affected ability to move limb but no explanation why. This would be very fruitful to find any related suggestion in sections Discussion and Conclusions. In addition, the last section is recommended to be enriched with outlook on the method used in the paper, its perspectives and implementation in clinical practice.
We sincerely appreciate the comment made. To be more specific on the effect of the chemotherapy on the patients, we have added the following sentence in the results section.
“In addition, chemotherapy is a treatment that uses powerful chemical drugs to kill cancer cells. However, it can produce side effects on the patients; most of the times it affects the physical condition, appearance, and the quality of life. Breast cancer patients undergoing adjuvant chemotherapy reduce their daily energy expenditure during therapy, which is associated with a loss of muscle mass. Furthermore, it was shown that skeletal muscle status is of clinical relevance because it is associated with treatment complications and time‐to‐tumor progression [28]. Therefore, a further analysis was performed among patients…”, lines 256-263.
[28] Klassen O, Schmidt ME, Ulrich CM, Schneeweiss A, Potthoff K, Steindorf K, Wiskemann J. Muscle strength in breast cancer patients receiving different treatment regimes. J Cachexia Sarcopenia Muscle. 2017; 8(2): 305-316. https://doi.org/10.1002/jcsm.12165
Furthermore, in the discussion section (lines 321-333), there is an explanation of why the chemotherapy treatment can affect the physical activity and the performance of the evaluated movements.
Finally, we have added the following paragraphs in the discussion section to emphasize the importance of our research in clinical practice.
“Furthermore, from the results found in the study, it can be recommended that the recovery process of the patients after breast cancer surgery should be supported by exercise therapy of the upper limbs.”, lines 347-349.
“The shoulder has been considered as one of the most complex joints of the upper limbs, so the understanding of its kinematics is difficult. Our research will contribute in an easy way to understand the relative movement of the arm with respect to the trunk, making practical the analysis of the patients who have this type of issues. In the clinical practice, our methodology will help to the physical rehabilitation team to understanding the recovery process of the patients with breast cancer.”, lines 350-355.
Remark 8.
English grammar and style are good but please double-check a couple of misprints.
Thank you very much; we have carefully reviewed the manuscript and corrected the missing grammar mistakes.

Reviewer 2 Report
In this study, authors evaluated the impact of mastectomy on the kinematics of the glenohumeral joint, using an optoelectronic motion capture system. As already reported in the limitations, the absence of details about the lymph node removal or the specific type of surgery could have influenced the results; however, it is interesting to have the same kinematic evaluation performed before and after surgery.
Overall, the article is well organized and structured, and results are clearly described in tables and figures.
Minor changes could improve the quality and the clarity of the study, especially in the section of Methods. The description of the study population should be improved with the inclusion of additional data, e.g. exclusion criteria, period of enrollment. Lines 68-69 and 73-74 should be moved to the Result section, eventually in a subsection 3.1 focused on the description of patients’ characteristics and summarized into a table. Moreover, authors should include BMI in the data instead of or in addition to mean body weight, considering the discussion about the presence of overweight women (lines 307-313).
Author Response
30th/March/2022
Dear Dr. Georgios Iatrakis
Guest Editor Healthcare
Manuscript ID: healthcare-1652495
Thanks for reviewing the manuscript entitled “Differences in the Glenohumeral Joint before and after Unilateral Breast Cancer Surgery: Motion Capture Analysis”. We sincerely appreciate the commentaries that allow us to improve the article.
We have carefully attended to each commentary made, so we proceed to send the manuscript again for review. The observations made and the responses by the authors are described below. We hope that the modifications made will achieve the final approval by the evaluators and the editorial committee, thanks in advance.
Note. The changes in the manuscript are marked up using the “Track Changes” function of the Microsoft Word program.
Kind Regards,
*************************************************************************
Reviewer 2
*************************************************************************
In this study, authors evaluated the impact of mastectomy on the kinematics of the glenohumeral joint, using an optoelectronic motion capture system. As already reported in the limitations, the absence of details about the lymph node removal or the specific type of surgery could have influenced the results; however, it is interesting to have the same kinematic evaluation performed before and after surgery.
Overall, the article is well organized and structured, and results are clearly described in tables and figures.
Minor changes could improve the quality and the clarity of the study, especially in the section of Methods. The description of the study population should be improved with the inclusion of additional data, e.g. exclusion criteria, period of enrollment. Lines 68-69 and 73-74 should be moved to the Result section, eventually in a subsection 3.1 focused on the description of patients’ characteristics and summarized into a table. Moreover, authors should include BMI in the data instead of or in addition to mean body weight, considering the discussion about the presence of overweight women (lines 307-313).
Thanks a lot for your commentaries and suggestions; these will help us to improve the quality of the manuscript. First, we have included a new subsection (2.1 Anthropometric data of the patients) with the anthropometric data, exclusion criteria, and some surgical characteristics of breast cancer. This information was summarized in Table 1 in the Materials and methods section as suggested. As this new section focused on the description of the patients, lines 68-69 and 73-74 were kept in the subsection.
Table 1. Anthropometric characteristics of the participants, and data related to breast cancer.
|
Anthropometric data |
Measurement Mean ± std |
Clinical data |
Measurement n (%) |
|
Age (years) |
46.7 ± 8.2 |
Stage cancer I and II |
8 (53.3%) |
|
Body weight (kg) |
68.4 ± 7.2 |
Stage cancer III |
7 (46.7%) |
|
Height (cm) |
156.9 ± 5.1 |
Chemotherapy (yes) |
8 (53.3%) |
|
Body Mass Index BMI (kg/m2) |
27.8 ± 3.1 |
Axillary lymph node dissection (yes) |
11 (73.3%) |
std = is the standard deviation of the mean; and n = represents the number of participants.
Exclusion criteria (Materials and Methods section:
“Women who did not completely heal the surgical wound before the 60 days, did not sign the informed consent and did not assist for the second evaluation after the surgery were excluded from the research.”, lines 97-99.
Thanks a lot for the commentary. We have added the mean and standard deviation of patients Body Mass Index in the subsection “2.1 Anthropometric data of the patients”. This with the purpose of being more specific about the description of the characteristics of the participants.
